# Management of Recurrence in Ovarian Cancer—The Role of Surgery and HIPEC with Relevance to BRCA Testing in a PARPi Landscape

**DOI:** 10.3390/cancers17040646

**Published:** 2025-02-14

**Authors:** Mathilde Duchon, Raj Naik, Fabrice Lecuru, Gwenaël Ferron, Caroline Cornou, Sabrina Madad Zadeh, Christophe Pomel

**Affiliations:** 1Department of Surgical Oncology, Centre Jean-Perrin, 58, rue Montalembert, 63011 Clermont-Ferrand, France; mathilde.duchon@clermont.unicancer.fr (M.D.); caroline.cornou@clermont.unicancer.fr (C.C.); sabrina.madad-zadeh@clermont.unicancer.fr (S.M.Z.); 2British Surgical Gynaecological Oncology Group (BSGOG), Newcastle upon Tyne NE2 4DJ, UK; naik@doctors.org.uk; 3Department of Surgical Oncology, Institut Curie, 26 Rue d’Ulm, 75005 Paris, France; fabrice.lecuru@curie.fr; 4Department of Surgical Oncology, IUCT Oncopole, Institut Universitaire du Cancer de Toulouse, 31100 Toulouse, France; ferron.gwenael@iuct-oncopole.fr

**Keywords:** recurrent ovarian cancer, surgical management, medical management, HIPEC

## Abstract

This review examines the surgical management of recurrent ovarian cancer, which remains complex and non-standardized. Analysis of five randomized trials (GOG-0213, DESKTOP III, SOC 1, HORSE, CHIPOR) highlights the importance of a personalized approach taking into account various factors including the timing of recurrence and the patient’s general condition, previous treatments and tumor/genetic characteristics. The results of the DESKTOP III and SOC 1 trials show a survival benefit for secondary surgery, whilst the GOG-0213 trial did not show an improvement in overall survival, underlining the need for careful patient selection using evidence-based selection criteria including the AGO or iMODEL scores. In patients with a negative score, there may be a place for cytoreductive surgery plus HIPEC in BRCA-negative cases following a course of chemotherapy, although current evidence shows no additional benefit of HIPEC when cytoreductive surgery is performed as an adjuvant procedure. This article provides a summary of the results of these major studies, whilst also addressing the issues that remain unresolved in the surgical management of recurrent ovarian cancer.

## 1. Introduction

Up to 70% of patients with advanced (stage III and IV) ovarian cancer will relapse within 3 years after the combination of surgery and platinum-based chemotherapy [1,2,3].

The surgical and medical management of recurrent ovarian cancer is complex and requires a personalized approach based on several factors, including the timing of the recurrence, the patient’s performance status, previous treatment regimens, the tumor’s histology, molecular characteristics, burden, and anatomy, and the ability of the patient to access expert centers for the treatment of ovarian cancer.

### 1.1. Understanding Recurrence in Ovarian Cancer

The GCIG—Gynecologic Cancer InterGroup—has defined several elements for assessing cancer recurrence: the CA-125 protein level, medical images, and clinical symptoms [4]. The GCIG considers that a recurrence can be suspected if the CA-125 level doubles in relation to its lowest reference value obtained during follow-up. Recurrence can be detected via CT, MRI, or PET scan. The appearance of clinical symptoms such as abdominal pain, bloating, or digestive problems may also indicate a recurrence, especially if the suspicion is confirmed by CA-125 measurement or imaging.

Recurrence in ovarian cancer is categorized based on the time interval from the completion of initial chemotherapy to the detection of relapse. Usually, classification is as follows: Platinum-Sensitive Recurrence, recurrence that occurs more than six months after the completion of platinum-based chemotherapy. Platinum-Resistant Recurrence: Recurrence that occurs within six months of completing platinum-based therapy. Platinum-Refractory Recurrence: Disease that progresses during platinum-based therapy. These definitions are now inadequate as sensitivity to platinum should be considered as a continuous variable, with response decreasing with early relapse. Therefore, the platinum-free interval (PFI) predicts the likelihood of response to relapse and has been used to define relapsed ovarian cancers rather than using the terms platinum sensitivity or resistance [5]. The Fourth Ovarian Cancer Consensus Conference of the Gynaecological Cancer Intergroup (GCIG) in 2010 adopted PFI as the defining criterion for enrolment in clinical trials for recurrent ovarian cancer. PFI is defined from the last date of platinum dose administration to documentation of disease progression. Accordingly, a PFI of less than six months is considered resistant or refractory disease, while a PFI of more than six months defines platinum-sensitive relapse [4].

To date, there are no data available concerning the impact on overall survival to support secondary cytoreductive surgery in platinum-resistant or refractory patients [6,7].

### 1.2. Therapeutic Strategies for Recurrent Ovarian Cancer

The management of recurrent ovarian cancer involves a range of therapeutic options, each tailored to the specific characteristics of the recurrence and the individual patient [1,8].

Platinum-Based Chemotherapy: Retreatment with a platinum-based regimen (such as carboplatin and paclitaxel or carboplatin and gemcitabine).PARP Inhibitors: Poly (ADP-ribose) polymerase (PARP) inhibitors, such as olaparib, niraparib, and rucaparib, have revolutionized the treatment of recurrent ovarian cancer, particularly in patients with BRCA mutations or homologous recombination deficiency (HRD). These agents have demonstrated significant efficacy in prolonging progression-free survival in both the platinum-sensitive and platinum-resistant settings.Anti-Angiogenic Agents: Bevacizumab, an anti-VEGF antibody, has been shown to improve progression-free survival in both platinum-sensitive and platinum-resistant recurrent ovarian cancer when added to chemotherapy. It can also be used as maintenance therapy following chemotherapy.Antibody drug conjugate: This treatment represents a promising advance in the treatment of ovarian cancer, particularly in platinum-resistant cases. Mirvetuximab soravtansine, an ADC targeting the folic acid receptor alpha (FRα), has recently been approved to treat platinum-resistant ovarian cancer, particularly in patients with high expression of FRα. The MIRASOL phase III trial showed a significant improvement in progression-free survival (PFS) and overall survival (OS) compared with standard chemotherapy [9]. Other ADCs under evaluation include luveltamab tazevibulin and raludotatug deruxtecan.Hormonal Therapy: Treatment with agents such as tamoxifen or aromatase inhibitors may be considered for patients with low-grade serous ovarian carcinoma or those who cannot tolerate chemotherapy. While the response rates are lower compared to chemotherapy or targeted therapies, hormonal agents are well-tolerated and can provide clinical benefit in selected cases.Surgery: Secondary cytoreductive surgery may be an option for some patients with recurrent ovarian cancer, particularly those with platinum-sensitive disease and a limited number of sites of recurrence. Several studies suggest that patients may benefit from complete secondary surgery.

To date, five randomized trials—GOG-0213 [10,11], DESKTOP III [12], SOC 1 [13,14,15], HORSE [16], and CHIPOR [17,18]—have been conducted and shed light on our practice.

## 2. GOG-0213—Secondary Surgical Cytoreduction for Recurrent Ovarian Cancer

This study was designed to explore two main questions. Firstly, the medical objective, i.e., the effect of adding bevacizumab to paclitaxel–carboplatin combination chemotherapy on overall survival (OS) and progression-free survival (PFS) in patients with recurrent platinum-sensitive ovarian cancer. Secondly, the surgical objective, i.e., investigating the role of cytoreductive surgery on overall survival.

This multicenter, open-label, randomized, phase 3 trial recruited patients with recurrent platinum-sensitive epithelial ovarian cancer. Inclusion criteria included ovarian cancer with a complete response to the first line of therapy, a chemotherapy-free interval of at least 6 months, a diagnosis of relapse (clinically evident relapse or biopsy confirmation). Exclusion criteria included more than one line of therapy, symptoms or diagnosis of gastrointestinal obstruction, grade 2 neuropathy, uncontrolled hypersensitivity to chemotherapy, and major surgery less than four weeks previously.

Patients meeting surgical eligibility criteria could participate in the surgical objective; this approach was predicated on the investigator-determined belief that recurrent disease could be completely resected before initiating chemotherapy.

Participants were randomized to receive standard chemotherapy with paclitaxel and carboplatin or standard chemotherapy plus bevacizumab. At the same time, patients eligible for secondary cytoreductive surgery were randomly assigned to undergo or not undergo surgery. The objective at surgery was the resection of tumor to no gross residual disease. The primary end point, overall survival, was measured from randomization to death from any cause on an intention-to-treat basis.

A total of 485 patients were randomly assigned to surgical cytoreduction followed by platinum-based chemotherapy (240 patients) or to chemotherapy alone (245 patients). Of 240 patients assigned to surgery, 225 (94%) eligible patients underwent secondary cytoreduction (per-protocol cohort) and 221 had operative reports confirming the postoperative tumor residual. The objective of complete resection was achieved in 63% of patients (150 of 239) in the cytoreductive surgery group. The median time to chemotherapy initiation in the surgery group was 40 days, as compared with 7 days in the no-surgery group. Most patients (408 of 485, 84%) received concomitant and maintenance bevacizumab. The use of each regimen was similar within each randomized cohort. The adjusted hazard ratio for disease progression or death (surgery vs. no surgery) was 0.82 (95% CI, 0.66 to 1.01) and the median progression-free survival was 18.9 months and 16.2 months, respectively. The percentage surviving without progression at 3 years was 29% (95% CI, 22 to 35) in the surgery group and 20% (95% CI, 15 to 26) in the no-surgery group. An exploratory ad hoc analysis was performed to assess the impact of complete gross resection on overall survival (OS) and progression-free survival (PFS) in the surgical population (239 patients). Adjustments were made for factors such as platinum-free interval, prior bevacizumab use, and the number of metastatic sites. Complete gross resection was associated with significantly longer OS (hazard ratio (HR) for death: 0.61; 95% confidence interval (CI): 0.40–0.93) and PFS (HR for disease progression or death: 0.51; 95% CI: 0.36–0.71) compared to incomplete resection. When comparing the complete resection subgroup (150 patients) to the entire no-surgery group (245 patients), no survival benefit was observed for OS (HR for death: 1.03; 95% CI: 0.74–1.46). However, there was a significant benefit in PFS for the resection group (HR for disease progression or death: 0.62; 95% CI: 0.48–0.80).

There was no significant difference between the surgical cytoreduction followed by platinum-based chemotherapy and chemotherapy alone groups in terms of overall survival or progression-free survival. Even when the analysis was limited to patients who had undergone complete macroscopic resection, surgery showed no advantage in terms of overall survival over chemotherapy alone. The median follow-up duration in this study is limited (48.1 months), which may restrict the ability to fully observe long-term effects or late outcomes.

## 3. DESKTOP III

The DESKTOP series (Descriptive Evaluation of Preoperative Selection Criteria for Operability in Recurrent Ovarian Cancer) is a trial series investigating surgery in patients with platinum-sensitive recurrence.

The DESKTOP I trial [19], using retrospective data, demonstrated the benefit of complete resection at primary debulking; only patients with complete resection in first-line treatment were associated with a long-term benefit for secondary debulking in recurrent ovarian cancer.

Consequently, a predictive score (the Arbeitsgemeinschaft Gynäkologische Onkologie (AGO)) was determined by the group to identify patients accessible to complete resection at the time of recurrence.

Multivariate analysis in the DESKTOP I trial identified the following independent predictive factors for achieving complete resection: complete resection during primary surgery, an Eastern Cooperative Oncology Group (ECOG) performance status of 0 (on a 5-point scale where higher scores indicate greater disability), and ascites ≤ 500 mL [20]. The AGO score was deemed positive when all three criteria were met. The DESKTOP II trial, a prospective, non-randomized, multicenter study, included 516 patients with platinum-sensitive relapse [21]. Among the 129 patients with a positive AGO score, complete resection was achieved in 76%. These findings validated the usefulness of the AGO score in predicting the likelihood of complete tumor resection.

The primary objective of the DESKTOP III trial was a randomized, multi-center study to determine whether secondary cytoreduction surgery followed by standard chemotherapy improves overall survival (OS) compared with chemotherapy alone in patients with recurrent platinum-sensitive ovarian cancer.

In terms of design, the study included patients with recurrent platinum-sensitive ovarian cancer (clinically defined as a lesion that is palpable or visible or that is visible on ultrasonographic imaging) who had limited recurrence and were considered operable (defined by a positive AGO score). Patients were randomized to secondary cytoreductive surgery followed by chemotherapy or chemotherapy alone. Randomization was stratified according to center and platinum-free interval (no previous chemotherapy, an interval of 6 to 12 months, or an interval of >12 months). The primary endpoint of the study was overall survival. The secondary endpoint was progression-free survival.

A total of 407 patients were enrolled. Macroscopic complete resection was achieved in 75.5% of the patients who were assigned to surgery and underwent the procedure. The median follow-up was 69.8 months. The median overall survival was 53.7 months (95% CI, 46.8 to 61.6) in the surgery group and 46.0 months (95% CI, 39.5 to 52.6) in the no surgery group (hazard ratio for death, 0.75; 95% CI, 0.59 to 0.96; *p* = 0.02). The median progression-free survival was 18.4 months (95% CI, 15.7 to 20.8) in the surgery group and 14.0 months (95% CI, 12.7 to 15.4) in the no-surgery group (hazard ratio for progression or death, 0.66; 95% CI, 0.54 to 0.82). The crossover rate (surgery with a subsequent recurrence) was 11%.

The results showed that patients who underwent secondary cytoreductive surgery had significantly longer overall survival than those who received chemotherapy alone. The median overall survival was 53.7 months in the surgery group versus 46.0 months in the no-surgery group (*p* = 0.02). Progression-free survival was also improved by 5.6 months (*p* < 0.001) in patients who underwent secondary cytoreductive surgery with complete resection compared with those who received chemotherapy alone. Although secondary cytoreductive surgery was associated with operative morbidity (surgical complications), these results support the policy that for patients selected according to AGO criteria, secondary surgery may offer a benefit in terms of overall survival.

DESKTOP III is the first randomized phase III trial to demonstrate an improvement in survival with secondary surgery for recurrent ovarian cancer. The benefit of secondary surgery was observed only in cases where a complete resection was achieved.

## 4. SOC 1

SOC 1 is also a randomized, multi-center phase 3 study that evaluated the efficacy of secondary cytoreductive surgery followed by chemotherapy versus chemotherapy alone in patients with recurrent platinum-sensitive ovarian cancer. The primary objective was to determine whether secondary cytoreductive surgery prior to chemotherapy improves progression-free survival compared to chemotherapy alone.

The study included patients with recurrent platinum-sensitive epithelial ovarian cancer. Patients had to have disease progression (with progression defined by RECIST (Response Evaluation Criteria In Solid Tumors) version 1.1) and a limited recurrence that could be completely resected. To identify patients eligible for complete surgery, the authors used the “iMODEL” based on several clinical and radiological factors that were assessed before surgery. The “iMODEL” score was calculated using six variables: International Federation of Gynecology and Obstetrics stage (FIGO), residual disease after primary surgery, platinum-free interval, performance status (measured by scales such as ECOG (Eastern Cooperative Oncology Group) or KPS (Karnofsky Performance Status)), serum level of cancer antigen 125 at recurrence, and the presence of ascites at recurrence. An iMODEL score of 4.7 or less out of 11.9 indicated a high probability of complete resection. A PET-CT scan was obtained during screening at each site, all images were assessed by two experienced, independent, nuclear medicine clinicians (blinded to treatment assignment). Patients were excluded if (1) it was deemed not possible to achieve complete resection according to their iMODEL score and PET-CT scan, (2) this was the second or later relapse, or (3) they previously had more than one line of chemotherapy. Patients were randomized into two groups: a group receiving secondary cytoreduction surgery followed by chemotherapy and a group receiving chemotherapy only without surgery. The primary endpoint was progression-free survival. The secondary endpoints included overall survival, quality of life, and treatment-associated complications.

A total of 357 patients were enrolled in this trial. Patients were randomly assigned to receive surgery (182 pts.) or control/no-surgery (175 pts.). The median progression-free survival was 17.4 months (95% CI 15.0–19.8) in the surgery group and 11.9 months (10.0–13.8) in the no surgery group (HR 0.58, 95% CI 0.45–0.74; *p* < 0.0001). The median overall survival was 58.1 months in the surgery group and 52,1 months in the no surgery group (HR 0.80, 95% CI 0.61–1.05, *p* = 0.109). Again, the benefit was restricted to patients with no macroscopic residual disease after surgery.

The study showed that patients who underwent secondary cytoreductive surgery followed by chemotherapy had significantly longer progression-free survival than those who received chemotherapy alone. In this study, 61/175 (35%) patients in the control group crossed over to surgery in subsequent therapies, justifying the adjustment of crossover. A sensitivity analysis showed that the adjusted HR for death for crossover to surgery was 0.76 (95% CI, 0.58 to 0.99). When excluding the center with the highest crossover rate, secondary cytoreduction provided a 21.0-month benefit to overall survival compared to the control/no-surgery group. Surgery did not increase OS in the intention-to-treat population but did result in a prolongation of survival following adjustment of crossover; median overall survival was 58.1 months in the surgery group and 49.5 months in the no-surgery group (hazard ratio [HR] 0.76; 95% CI 0.58–0.99).

These results support the use of secondary cytoreductive surgery in the management of recurrent ovarian cancer in selected patients.

## 5. HORSE

The HORSE trial (Hyperthermic Intraperitoneal Chemotherapy in Platinum-Sensitive Recurrent Ovarian Cancer: A Randomized Trial on Survival Evaluation) was recently published in the *Journal of Clinical Oncology* [16]. This multicenter, randomized phase III study aimed to evaluate whether adding hyperthermic intraperitoneal chemotherapy (HIPEC) to secondary cytoreductive surgery, in the absence of neoadjuvant chemotherapy, improves progression-free survival compared to secondary cytoreductive surgery alone.

All patients underwent surgery, and patients were randomized during surgery if their residual tumor was ≤0.25 cm. In the experimental arm, HIPEC with cisplatin at 75 mg/m^2^ over 60 min at 41.5 °C was administered at the conclusion of the operation. Both groups received postoperative platinum-based chemotherapy. The primary outcome was progression-free survival, while the secondary outcomes included safety profiles and recurrence survival.

The study included 167 patients, with 82 in the secondary cytoreductive surgery plus HIPEC group (SCS plus HIPEC group) and 85 in the secondary cytoreductive surgery-alone group (SCS-alone group). After a median follow-up of 83 months (IQR 64-102), the median PFS was 23 months (95% CI, 17–29) in the SCS-alone group and 25 months (95% CI, 18–32) in the SCS plus HIPEC group. There was a difference of 2 months in favor of the SCS + HIPEC group, but this was not statistically significant. The 5-year probability of RS was 61.6% (95% CI, 50.8–72.4) in the SCS group and 75.9% (95% CI, 66.5–85.3) in the SCS plus HIPEC group. Although the SCS + HIPEC group showed better results, this is still insufficient to conclude that there is any real significant clinical benefit. The rates of postoperative adverse events were similar across both groups, suggesting that the addition of HIPEC does not increase the risk of postoperative complications.

The addition of HIPEC to primary complete or near-complete cytoreduction did not provide a significant benefit in terms of progression-free survival in patients with platinum-sensitive peritoneal recurrence. In addition, the use of HIPEC is well tolerated, with a safety profile comparable to that of surgery alone. When examining details of the surgery, in 95% of cases resection was complete and in 5% tumor residue was <0.25 cm. For most patients (87%), the complexity of the surgery was described as simple or intermediate (according to the Surgical Complexity Score 1–2). When analyzing the genetic characteristics of the population, 43.5% of patients had a BRCA mutation and favored a longer survival [22,23]. This is therefore a highly selected population, thereby explaining a prolonged PFS of 24 months.

## 6. CHIPOR

The results of the CHIPOR trial (HIPEC in platinum-sensitive relapsed epithelial ovarian cancer) were presented at ASCO in 2023 [18]. The primary objective of the CHIPOR trial was to determine whether the addition of hyperthermic intraperitoneal chemotherapy (HIPEC) after secondary cytoreductive surgery would improve progression-free survival compared with standard intravenous chemotherapy followed by surgery alone in patients with recurrent platinum-sensitive ovarian cancer.

The study, conducted in 31 institutions, included patients with recurrent platinum-sensitive epithelial ovarian cancer who were candidates for secondary cytoreductive surgery with a goal of complete tumor resection.

Patients received six cycles of platinum and taxane chemotherapy combined with bevacizumab. Those eligible for complete cytoreductive surgery at the end of chemotherapy were enrolled and randomly assigned at surgery to either undergo HIPEC (cisplatin 75 mg/m^2^ at 41 °C for 60 min) or no additional treatment. Randomization was conducted after achieving complete cytoreduction and stratified based on the treatment center, surgical outcome (no residual disease vs. residual disease ≤0.25 cm), chemotherapy-free interval before relapse, and prior use of PARP inhibitors (yes vs. no).

The primary endpoint of the study was overall survival (OS). The target sample size was set at 404 evaluable patients, with 80% statistical power at a 5% significance level after 268 deaths. Secondary endpoints included progression-free survival (PFS), peritoneal PFS, patient-reported outcomes, safety, and postoperative morbidity and mortality (within 60 days post-surgery).

A total of 415 patients were randomized between 2011 and 2021. Baseline characteristics were balanced between treatment arms. At the data cutoff, with a median follow-up of 6.2 years, 268 patients (65%) had died. Adding HIPEC to cytoreductive surgery after six cycles of second-line chemotherapy for patients with first late relapse of ovarian cancer significantly improved overall survival, 54.3 months (41.9–61.7) versus 45.8 months (38.9–54.2) in the no-HIPEC arm (HR: 0.73 [95% CI 0.56–0.96], *p* = 0.024). Progression-free survival was also improved in the HIPEC arm, 10.2 months (9.3–11.9) versus 9.5 months (8.6–11.6) in the no-HIPEC arm (HR: 0.79 [95% CI 0.63–0.99]). The time to start subsequent therapy was also significantly improved with HIPEC.

Regarding maintenance treatment, 11% of patients in the HIPEC arm received bevacizumab, compared with 16% in the no-HIPEC arm. BRCA mutation status was known in 85% of patients in the HIPEC arm and in the no-HIPEC arm; 24% of patients had a mutation in the HIPEC arm versus 25% in the no-HIPEC arm, and PARP inhibitor maintenance treatment was initiated following surgery in 21% of patients in the HIPEC arm versus 20% in the no-HIPEC arm. In the subgroup with BRCA mutation (of whom 70% received a PARP inhibitor), overall survival and progression-free survival (global, peritoneal, and extraperitoneal) appeared to favor the no-HIPEC group.

In the face of changing clinical practice, post hoc analyses examined the differences between patients recruited in the first 5 years of the trial (2011–2016) and those recruited later (2017–2021). In line with changing standards of care, 57% of patients were receiving a PARP inhibitor in the later period compared with 26% in the earlier period. However, there was no difference in overall survival and progression-free survival between both time-periods [17].

Grade 3 morbidity according to the Clavien–Dindo classification was 35% in the HIPEC group compared with 17% in the surgery alone group. The majority of grade III toxicities in the HIPEC group were renal toxicities due to the lack of use of Natrium Thiosulfate for prevention during the first part of the study. There was also a stoma rate of 9.7% in the HIPEC group compared with 4.8% in the surgery alone group. The median duration of surgery was 119 min longer in the HIPEC group.

This study is the first large-scale study of HIPEC in recurrent disease to show a benefit on overall survival. However, CHIPOR does not answer the question of whether or not surgery is beneficial, nor does it validate surgery after chemotherapy at the time of relapse, as surgery after six cycles of chemotherapy is not currently considered standard practice either in primary presentation or at recurrence.

## 7. Discussion

Although both the DESKTOP III and the SOC 1 trials support the benefit of secondary surgery, the GOG-0213 trial did not show an overall survival benefit. The results of GOG-0213 confirmed that surgery should not be offered to all patients with platinum- sensitive recurrent ovarian cancer and highlighted the importance of strict patient selection using evidence-based selection criteria including the AGO and iMODEL scores [24].

Also, in the two trials showing a benefit from surgery—DESKTOP III and SOC 1—the benefit was present and observed exclusively in patients where complete resection was achieved, underlining the importance of selecting the right patients and utilizing the right surgical teams.

### 7.1. The Populations Are Different

When we more closely analyze the populations in each trial, we can observe significant and relevant differences. Table 1 compares the characteristics of each trial population.

Firstly, the enrolled patients in SOC-1 were relatively younger than those enrolled in DESKTOP III and GOG-0213. The median platinum-free interval is also higher in the DESKTOP trial.

When we investigate the complete resection rate, it is lowest in the GOG-0213 study, with a rate of 63%. In this study, patient selection was solely based on the surgeon’s judgement. In the DESKTOP III and SOC 1 studies, the use of a patient selection score resulted in a complete resection rate of >75%. DESKTOP III selected patients using the AGO criteria and SOC-1 selected patients using the iMODEL score combined with PET-CT. It is also worth noting that the iMODEL score combined with PET-CT in SOC-1 selected more potential candidates than the AGO score in DESKTOP III. By comparison, the AGO score was more restrictive and only included recurrent patients who had achieved complete resection at primary surgery [14]. In the DESKTOP II study, 512 patients with platinum-sensitive relapse were selected, and 261 patients (51%) had a positive AGO score [12]. The AGO score eliminated almost 49% of patients with platinum-sensitive relapse. In the CHIPOR trial, 84.5% of patients had a positive AGO score [17].

### 7.2. The Strategy Is Different

The therapeutic strategies of these four trials differ. The CHIPOR trial showed a benefit to HIPEC after six cycles of neoadjuvant chemotherapy in patients with complete resection. This involves starting with platinum-based chemotherapy and then performing surgery with HIPEC. The HORSE study showed that the addition of HIPEC to secondary cytoreductive surgery, without prior neoadjuvant chemotherapy, did not significantly improve progression-free survival compared with surgery alone in platinum-sensitive recurrences. DESKTOP, SOC I, and GOG-0213 involve starting with cytoreduction surgery followed by chemotherapy.

The management of recurrent ovarian cancer remains a significant surgical challenge, requiring a multidisciplinary approach and a personalized treatment strategy.

Should recurrent ovarian cancer be operated on? Yes, if surgery is performed before chemotherapy and only when using the AGO/iMODEL score and in expert centers with expert surgeons and without the addition of HIPEC.Regarding the role of HIPEC, it can be proposed in BRCA-negative cases when cytoreductive surgery is performed following response after neo-adjuvant chemotherapy in cases who have an initial negative AGO and iMODEL score. There appears to be no benefit of HIPEC in BRCA-positive cases in this setting and the current recommendations are they be treated with PARP inhibitors.Figure 1 shows an algorithm for surgery in the case of recurrent disease, based on current data.

### 7.3. The Grey Areas

When we look at the histological type of these studies, more than 80% were high-grade serous cancers, making it difficult to generalize the conclusions of these articles to other histological types.

There are many unanswered questions concerning the results of these four studies, taking into consideration the new adjuvant therapies, anti-PARP therapies, and bevacizumab, which are now routinely used. In the GOC-0213 study and DESKTOP study, we have no knowledge of the BRCA and HRD status; in the SOC-1 study this information was missing in 79% of cases. In the CHIPOR trial, the BRCA status was known in 85% of patients, 24.5% had a mutation and 20.5% received anti-Parp therapy. In these patients who received adjuvant anti-Parp treatment, the results of CHIPOR did not seem to show any benefit from HIPEC. In the HORSE study, the results of the sub-group analysis showed no benefit to HIPEC in BRCAm patients. In the OVHIPEC-1 trial, patients with homologous recombination deficiency (HRD) without a pathogenic BRCA1/2 mutation benefited more from HIPEC in terms of overall and progression-free survival. The results for patients with BRCA1/2 mutations were less clear, suggesting a potentially lesser impact for this sub-population [25,26]. Koole et al. showed that patients with HRD tumors without pathogenic BRCA1/2 mutation appear to benefit most from treatment with HIPEC [27]. The currently changing situation, with the loss of platinum sensitivity in the event of recurrence on iPARP, raises questions about the probable need to prove sensitivity to chemotherapy before secondary surgery. Future studies on surgery for recurrent disease should focus on the following:Surgery following neoadjuvant chemotherapy in patients with a negative AGO and negative iMODEL/PET-CT score stratified by BRCA/HRD status and subsequent PARPi therapy;HIPEC as adjuvant therapy prior to chemotherapy stratified by BRCA/HRD status and subsequent PARPi therapy;Development of a new score/model in patients previously treated by PARPi stratified by BRCA/HRD status.

The TORPEDO trial (Trial of Optimal Radical Cytoreductive Surgery Plus or Minus HIPEC in Ovarian Cancer) showed that subtotal peritonectomy after neoadjuvant chemotherapy resulted in excellent progression-free survival and a very low rate of platinum-resistant recurrence [28]. If a patient has had multiple previous surgeries, particularly extensive surgeries for ovarian cancer with extended peritonectomies, the benefits of additional surgery may diminish. Scar tissue (adhesions) and other complications can make surgery more difficult and less likely to be successful. The site of recurrence also needs to be taken into account, especially when bowel resections were performed during the initial surgery. The risk of a permanent stoma must also have been clearly discussed and accepted by the patient. It is also uncertain whether patients with resectable intra-abdominal recurrence but with supra-diaphragmatic adenopathy should be operated on. If the disease is widespread or difficult to remove surgically, the risks may outweigh the potential benefits. For many patients, complete cytoreduction surgery is not feasible. In these situations, when cytoreductive surgery is not an option, medical treatment is preferred, although some patients may require palliative surgery if the recurrent disease is causing significant symptoms.

## 8. Conclusions

The decision to perform surgery for recurrent ovarian cancer depends on several factors, including the individual patient’s condition, the characteristics of the recurrence, the molecular and genetic status of the tumor, and the overall treatment goals. By understanding the nuances of each patient’s disease and utilizing the latest advancements in medical and surgical treatment, healthcare providers can optimize care and improve survival and quality of life for women facing recurrent ovarian cancer.

## Figures and Tables

**Figure 1 cancers-17-00646-f001:**
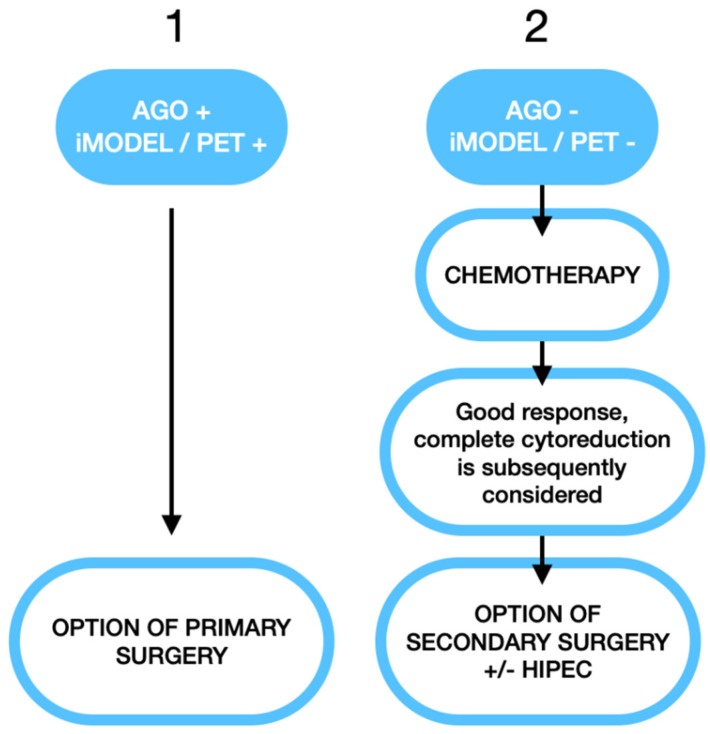
Algorithm for surgery in the case of recurrent disease.

**Table 1 cancers-17-00646-t001:** Comparison of populations from the GOG-0213, DESKTOP III, SOC 1, and CHIPOR trials.

	GOG-0213	Desktop III	SOC 1	CHIPOR	HORSE
Age (year)	60.6	60.58	54.2	60.5	55
Initial stage III-IV	85%	78.2%	82%	86%	66%
Selection criteria	Individualized	AGO Score	iMODEL + PET-CT	Complete intra-operative resection and platinum sensitivity	Complete intra-operative resection and NOT receive chemotherapy before surgery
Histology: high grade serous	85%	80.6%	85%	74.5%	75%
Median platinum-free interval (month)	18.8	21.1	16.1	17.6	18
Previous platinum-free interval	6–12 months 25% >12 months 75%	6–12 months 25% >12 months 75%	6–16 months 46% >16 months 54%	6–12 months 26% >12 months 74%	/ /
Moment of randomization	Before treatment	Before treatment	Before treatment	After chemotherapy, during surgery	During surgery
Median Follow-up (month)	48.1	69.8	82.5	74	83
Complete gross resection	63%	75.5%	76.7%	87%	75.5%
Crossover rate (surgery with a subsequent recurrence)	Unknown	11%	36,9%	/	/
Mortality	30 days: 0.4%	90 days: 0.5%	60 days: 0%	30 days: 0.7%	90 days: 0
Platinum-based combination therapy	82%	89%	?	100%	100%
2nd-line bevacizumab	84%	23%	1%	13.5%	11.4%
2nd-line PARPi maintenance	NA	<5%	10%	20.5%	<5%

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
