# Peer review of "Management of Recurrence in Ovarian Cancer—The Role of Surgery and HIPEC with Relevance to BRCA Testing in a PARPi Landscape"

_cancers, 2025, doi:10.3390/cancers17040646_

Round 1
Reviewer 1 Report
Comments and Suggestions for Authors
Authors compared four large trials, GOG0213, DESKTOP III, SOC1, and CHIPOR, and described them in detail. They provide a good summary of the evidence for surgical therapy in recurrent ovarian cancer, but all of this is known and lacks novelty.
As a Reviewer, I would make a few comments.
P2L64: Authors describe treatment options as “Therapeutic strategies for Recurrent Ovarian Cancer”. The indications vary from patient to patient, and the order of priority in selecting these options should be described first.
While the eligibility criteria and representative scoring systems for each clinical trial are described, the authors should also mention an overview of the indications for surgery in general practice.
The grey areas in the Discussion are the most interesting and important. It would be desirable to organize a list of unresolved issues as well as evidence that should be considered in the future.
Author Response
We agree with reviewer 1.
We emphasized in the revised version the need for selection of patients particularly with BRCA status.
Reviewer 2 Report
Comments and Suggestions for Authors
The authors are addressing an important question about the role of surgical management in patients with platinum sensitive ovarian cancer recurrence. For this, they complied a summary of 4 RCT's.
Comments:
1.Title:
Management of Recurrence in Ovarian Cancer—Place of the Surgery
Please consider re-titling your paper. “The Role of Surgery in the Management of Recurrent Ovarian Cancer” or something similar is more suitable. “place of the surgery” suggests the paper will be discussing the implications of where the surgery took place.
Abstract:
Requires proof read
Methods are missing in the abstract
Intro:
Please add citations line 52 and 54
Methods?
Results?
99-103: a bit difficult to understand. Please phrase more clearly, in concordance with the objectives as defined in the study itself
172: first relapse
196: worth mentioning that patients in the surgical arm that did not achieve CGR at secondary cytoreductive surgery had worse prognosis than patients that had chemotherapy alone
256: please clarify ?
298: consider adding a Kaplan Meier OS of the 4 studies to enable visual demonstration of the survival differences
Discussion:
Line 327:
consider discussing the correlation between those findings to explain the difference in the outcome of the 3 studies. for example, as OS in surgical group in gog213 and desktopIII was very similar (53.6 vs 53.7 months respectively) whereas the OS in the non surgical group of GOG213 was 65.7, the addition of bev to the regimen should be discussed as driving factors. also, surgical trials are difficult to conduct and interpret as quality assurance of the surgical procedure is difficult to assess, etc.
might be worth mentioning other RCT's that studies the role of anti angiogenesis in recurrent ovarian cancer (ICON6 , OCEANS, MITO16b, AGO-OVAR) as this can be relevant to explain the discrepancy between GOG213 and desktop.
Might be worth mentioning a systemic review and meta-analysis of those trials that was published in 2021 (Ding T, Tang D, Xi M. The survival outcome and complication of secondary cytoreductive surgery plus chemotherapy in recurrent ovarian cancer: a systematic review and meta-analysis. J Ovarian Res. 2021 Jul 13;14(1):93. doi: 10.1186/s13048-021-00842-9. PMID: 34256813; PMCID: PMC8278673.)
Author Response
The manuscript has been reviewed by a native english writer.
We retitled the paper by emphasized the molecular consideration.
Citations has been adding
99 to 103 has been totally rephrased
As we add the HORSE study, Populations of studes are different and statistician do not consider to use kaplan OS of the five studies.
The addition of bev to the regimen has been discussed.
We do agree that surgical trials are difficult to conduct and interpret as quality assurance of the surgical procedure is difficult to assess.
The a systemic review and meta-analysis of those trials that was published in 2021 (Ding T, Tang D, Xi M. ) did not take into consideration the HORSE trial and BRCA mutation.
Reviewer 3 Report
Comments and Suggestions for Authors
My major point of concern is the novelty of the present work. In 2023, Fotopoulou et al. published their work on the exact same topic:
Fotopoulou C, Eriksson AG, Yagel I, Chang SJ, Lim MC. Surgery for Recurrent Epithelial Ovarian Cancer. Curr Oncol Rep. 2024 Jan;26(1):46-54. doi: 10.1007/s11912-023-01480-8.
The only study missing would be CHIPOR, however, I am afraid that the addition of this study does not justify the publication of a novel narrative review as a summary of these studies.
Author Response
Please note that the publication of
Fotopoulou C, Eriksson AG, Yagel I, Chang SJ, Lim MC. Surgery for Recurrent Epithelial Ovarian Cancer. Curr Oncol Rep. 2024 Jan;26(1):46-54. doi: 10.1007/s11912-023-01480-8.
did not consider both CHIPOR and HORSE trial. THis is extremely relevant at the era of HIPEC.
Also, the BRCA mutation patients is now extremely relevant for selection of patients.
Round 2
Reviewer 1 Report
Comments and Suggestions for Authors
I confirm that Authors have responded sincerely to the Review comments. In addition, I believe that the revised version was a useful Review with the addition of the HORSE Study.
The title of the report, as well as the content, was designed to discuss the issues and Unmet Needs that emerge in light of the evidence.